# Spatiotemporal Evolution of Ground Subsidence and Extensional Basin Bedrock Organization: An Application of Multitemporal Multi-Satellite SAR Interferometry

Carlo Alberto Brunori [1,*] and Federica Murgia [2]

1    Istituto Nazionale di Geofisica e Vulcanologia, 52100 Arezzo, Italy
2    CREA-Centro Foreste e Legno, 38123 Trento, Italy
*    Correspondence: carloalberto.brunori@ingv.it

**Abstract:** Since the early 1990s, the European (ESA) and Italian (ASI) space agencies have managed and distributed a huge amount of satellite-recorded SAR data to the research community and private industries. Moreover, the availability of advanced cloud computing services implementing different multi-temporal SAR interferometry techniques allows the generation of deformation time series from massive SAR images. We exploit the information provided by a large PS dataset to determine the temporal trend of ground deformation and the relative deformation rate with millimetric accuracy to analyze the spatial and temporal distribution of land subsidence induced by water pumping from a deep confined aquifer in the Northern Valle Umbra Basin (Central Italy), exploiting 24 years of Permanent Scatterers—interferometric SAR data archives. The SAR images were acquired between 1992 and 2016 by satellites ERS1/2 and ENVISAT, the Sentinel 1 ESA missions and the COSMO-SkyMed ASI mission. We observed ground velocities and deformation geometries between 1992 and 2016, with displacements of more than 70 cm and velocities of up to 55 mm/yr. The results suggest that the shape and position of the surface ground displacement are controlled by the fault activity hidden under the valley deposits.

**Keywords:** subsidence; time series; PS; InSAR; Valle Umbra Basin; buried faults

## 1. Introduction

In the last few decades, Earth's observation missions have produced a huge volume of data acquired by active and passive sensors operating in several regions of the electromagnetic spectrum with territorial continuity and very high spatial and temporal resolutions [1]. In this framework, the long time-series of Synthetic Aperture Radar (SAR) image catalogues freely available by the national and international space agencies (such as the European and Italian Space Agencies, ESA and ASI, respectively) can be processed using Multi-temporal Interferometric Synthetic Aperture Radar (MInSAR) techniques, in some cases using cloud computing implementation of processing chains [1–3], to produce the measure of ground displacements with high temporal and geometric resolution also at the national scale [4]. The increasing availability of time series produced in the framework of research and application projects implementing MInSAR procedures or online services for the generation of land displacement measures allows the monitoring of natural or anthropogenic phenomena in their evolution.

One of the applications of MInSAR satellite techniques is monitoring ground subsidences [5] affecting populated areas such as flat valleys with deformation rates of a few millimeters up to decimeters for a year. The subsidences are mainly related to several processes, often in combination with each other, such as the aquifer-system compaction due to natural processes or the artificial groundwater extraction for agricultural and industrial purposes. In certain cases, rapid urbanization can affect unconsolidated alluvial or basin-fill aquifer systems. Indeed, when groundwater migrates from aquifers, the pressure in the

water pore decreases, causing the effective stress in the soils and rocks to increase and induce gradual land subsidence as a consequence of the consolidation of the aquifers [6]. Using MInSAR techniques, surface ground subsidence phenomena have been observed and attributed to the compaction of sediments in places such as London [7], Shanghai [8], Mexico City and other cities in Central Mexico [9–11]. In Italy, some areas are affected by ground subsidence, such as Roma [12], Bologna town and surroundings [13], the Ravenna coastal plain [14] or Italian coastlines [15], the whole Po river delta [16], the Venezia lagoon [17], the Arno river basin [18], the Gioia Tauro [19] and the Sibari coastal plains [20]. In some cases, these deformations can damage urban infrastructures, such as water, gas and electric utilities, as well as buildings and roads [21]. Subsidences have also been observed in areas with dissolution in the subsoil of soluble rocks, such as limestone, dolomite and gypsum, or as a consequence of mining [21,22]. Moreover, several studies on flat valleys have tried to identify evidence of fault systems dislocating the bedrock buried under the valley deposits, such as lateral variations in ground deformation velocity [23,24].

　　　To demonstrate the use of MInSAR time series products, this work analyzes the slow vertical ground deformations (subsidences) that occurred in a wide territory of Central Italy [25], the Northern Valle Umbra Basin (NUB), between 1992 and 2016 by exploiting two Permanent Scatterers (PS) datasets derived from 24 years (1992–2016) of SAR data (Figure 1). Four satellite missions acquired the SAR data employed for the NUB subsidence analysis: ERS 1-2 (ERS), ENVISAT (ENV), COSMO-SkyMed (CSK) and Sentinel 1 (SNT) (Figure 1a,b, Table 1).

**Table 1.** Surface statistics for the clusters of the four satellite missions of the MATTM database (ERS, ENV and CSK) and the MEMPHYS database (SNT).

|  | ERS | | ENV | | CSK | | SNT | |
|---|---|---|---|---|---|---|---|---|
|  | ASC | DESC | ASC | DESC | ASC | DESC | ASC | DESC |
| Numb of PS | 4.378.310 | 9.438.651 | 15.359.988 | 12.895.230 | 65.019.955 | 63.735.666 | 611.986 | 721.982 |
| Numb. of Clusters | 117 | 201 | 110 | 95 | 56 | 56 | 1 | 1 |
| PS/kmq (avg.) | 2.150,1 | 5.905,2 | 5.344,1 | 4.495,3 | 47.774,7 | 46.447,8 | 51.58 | 60.85 |
| Max | 218.468 | 264.282 | 747.622 | 756.312 | 4.601.190 | 4.242.770 |  |  |
| Min | 881 | 518 | 205 | 258 | 20.969 | 4.940 |  |  |
| Mean | 37.744 | 47.193 | 139.635 | 135.739 | 1.154.140 | 1.051.230 |  |  |
| Median | 27.902 | 34.248 | 72.690 | 70.995 | 872.943 | 893.517 |  |  |

　　　In the following, after an overview of the geological settings of the study area and a presentation and quality check of the exploited PS datasets, ground deformation velocity maps, both derived from the above-mentioned PS datasets, are shown. Therefore, the use of large PS datasets to analyze the relation between ground subsidence and the variation of the piezometric level in the wells extracting water from a deep-confined aquifer will be described and discussed. Moreover, the possible correlation between the location of surface vertical displacement and valley morphology and the presence of faults that dissect the bedrock under the valley deposits will be analyzed.

　　　One of the objectives of this work is the analysis of the already processed and available PS time-series provided by two projects (PST Italian Ministry and MEMPHIS EU Project). This kind of dataset widely circulates within the scientific community, and we believe it is useful to understand their full potential by using them for various purposes.

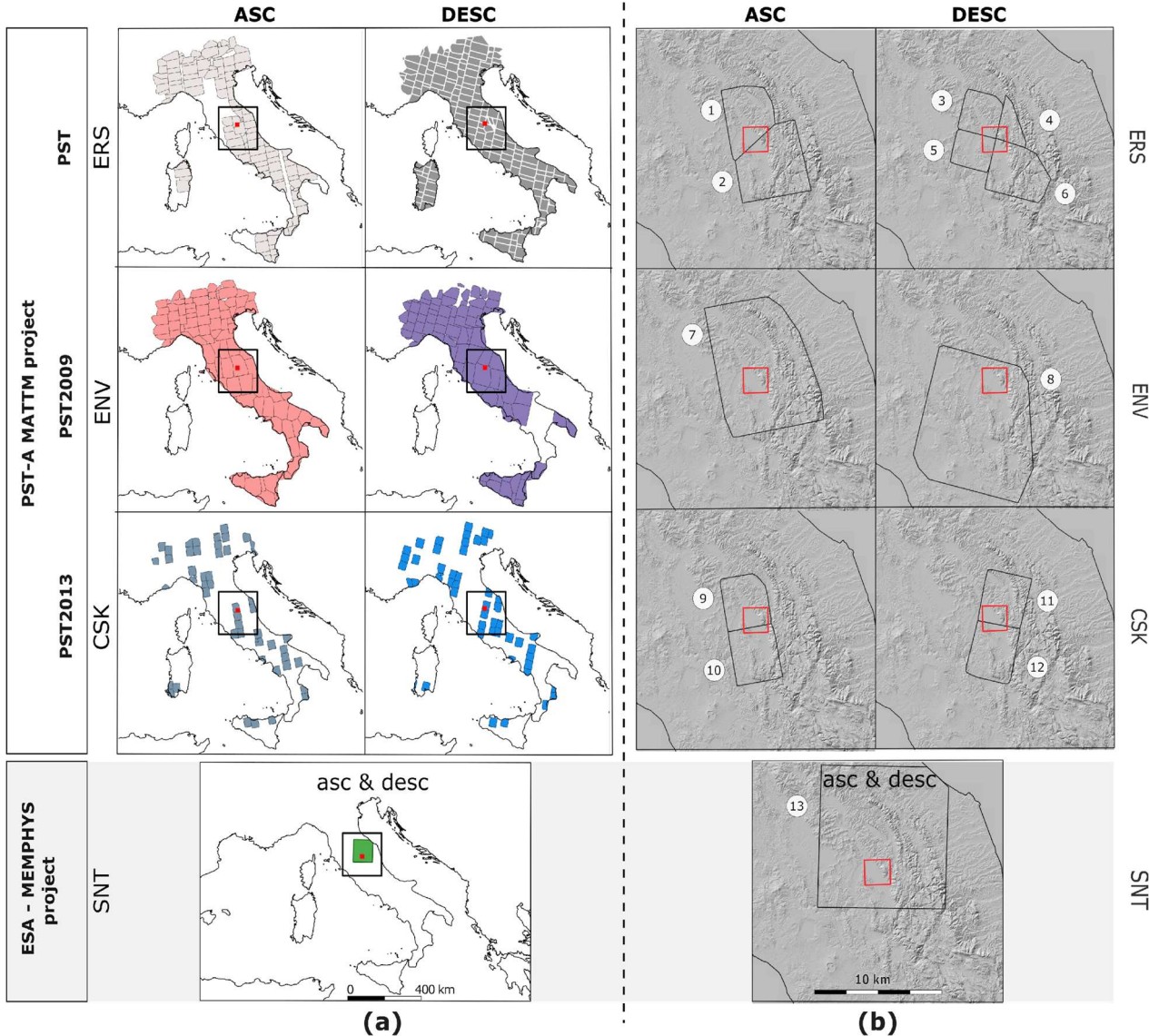

**Figure 1.** (**a**) The PS data on the Italian peninsula provided by the PST-A project (the first three rows are the data from the 1992–2014 ERS, ENV and CSK missions) were organized in clusters of ascending (ASC) and descending (DESC) orbits. In the last row, the Italian area covered by PS is derived from the SNT mission in both ascending and descending satellite orbit data, processed by the ESA-MEMPHIS project. See Table 1 for the statistics. The black box indicates the (**b**) area. (**b**) The red box indicates the NUB study area, and the numbers in the white circles identify the clusters' names (Tables 2 and 3) used in this work. In the background, the central Italy orography and black boxes are the clusters of PS used in this work.

**Table 2.** The 1993–2016 PS dataset covering the study area derives from SAR data acquired by four satellites carrying sensors operating in the C (ERS, ENV and SNT) and X bands (CSK). The dates are expressed in the dd/mm/yyyy format.

|  | ERS 1-2 | ENV | CSK | SNT1 |
|---|---|---|---|---|
| Band (wavelen. cm) | C-band (5.6) | C-band (5.6) | X-Band (3.1) | C-band (5.6) |
| Operation mode | SAR/IM | SAR/IM | HIMAGE (Stripmap) | TOPSAR |
| Revisit cycle days | 35 | 35 | 16 | 12 |
| Look angle | 23° | 23° | 25–57° | 30.5° |
| Swath km | 100 | 100 | 40 | 250 |
| ASC n. images | 35 | 51 | 40 | 48 |
| ASC first image | 2 April 1995 | 2 December 2002 | 9 May 2011 | 25 October 2014 |
| ASC last image | 23 October 2000 | 24 May 2010 | 30 March 2014 | 21 August 2016 |
| DESC n. images | 56 | 37 | 30 | 42 |
| DESC first image | 21 April 1992 | 10 October 2003 | 29 July 2011 | 24 October 2014 |
| DESC last image | 29 December 2000 | 25 June 2010 | 16 April 2014 | 1 September 2016 |

**Table 3.** The PS dataset includes the satellite missions, orbits, project identification code, project identification file name and list of PS clusters used for our analysis of the NUB area (see Figure 1b).

|  | Orbit | Project ID | Cluster ID | Number in Figure 1b |
|---|---|---|---|---|
| ERS 1-2 | ASC | PST | ERS_T401_F858_CL003_SPELLO | 1 |
|  |  |  | ERS_T401_F858_CL004_SPOLETO | 2 |
|  |  |  | ERS_T351_F2745_CL002_PERUGIA | 3 |
|  | DESC | PST | ERS_T79_F2748_CL001_GUALDO | 4 |
|  |  |  | ERS_T79_F2748_CL003_SPOLETO | 5 |
|  |  |  | ERS_T351_F2745_CL001_MARSCIANO | 6 |
| ENV | ASC | PST2009 | ENVISAT_T401_F858_CL001_ASSISI | 7 |
|  | DESC | PST2009 | ENVISAT_T351_F2745_CL001_TERNI | 8 |
| CSK | ASC | PST2013 | CSK_F_44_PERUGIA_A_CL001 | 9 |
|  |  |  | CSK_F_43_SPOLETO_A_CL001 | 10 |
|  | DESC | PST2013 | CSK_F_46_SPOLETO_D_CL001 | 11 |
|  |  |  | CSK_F_45_GUALDOTADINO_D_CL001 | 12 |
| SNT | ASC | SqueeSAR | S1_T117_A_33 | 13 |
|  | DESC | SqueeSAR | S1_T95_D_32 | 13 |

## 2. Case Study

Ground subsidence occurring in the northern sector of the NUB [25] is measured using PS data generated by exploiting MInSAR techniques. The NUB (also called "Spoletana" Valley) is an NNW-SSE 20 km long and 10 km wide flat valley located west of the "inner ridge" of the Northern-Central Italian Apennines [26–29] (Figure 2). The local highest elevation is 1290 m of Mt. Subasio, and the valley has elevations between 170 m.a.s.l. near the northern confluence with the Tevere Valley and 240 m.a.s.l. on the southern side, in Foligno town (on the Topino River conoid, Figure 2). The principal cities in the area are Perugia and Foligno. Perugia, with about 163,000 inhabitants, is the capital city of both the Umbria Region and the homonymous province. It is located to the right of the Tevere River Valley at the junction with the NUB at the northernmost limit. Foligno (55,300 inhabitants) is located at the southernmost limit of the studied area. The valley is populated with intense agricultural activities and several industrial settlements. Principal towns and villages of the NUB are, from north to south: Santa Maria Degli Angeli, Assisi, Tordandrea, Cannara, Bastia, Foligno, Spello, Bevagna and Foligno (Perugia Province, Umbria Region, Figure 2).

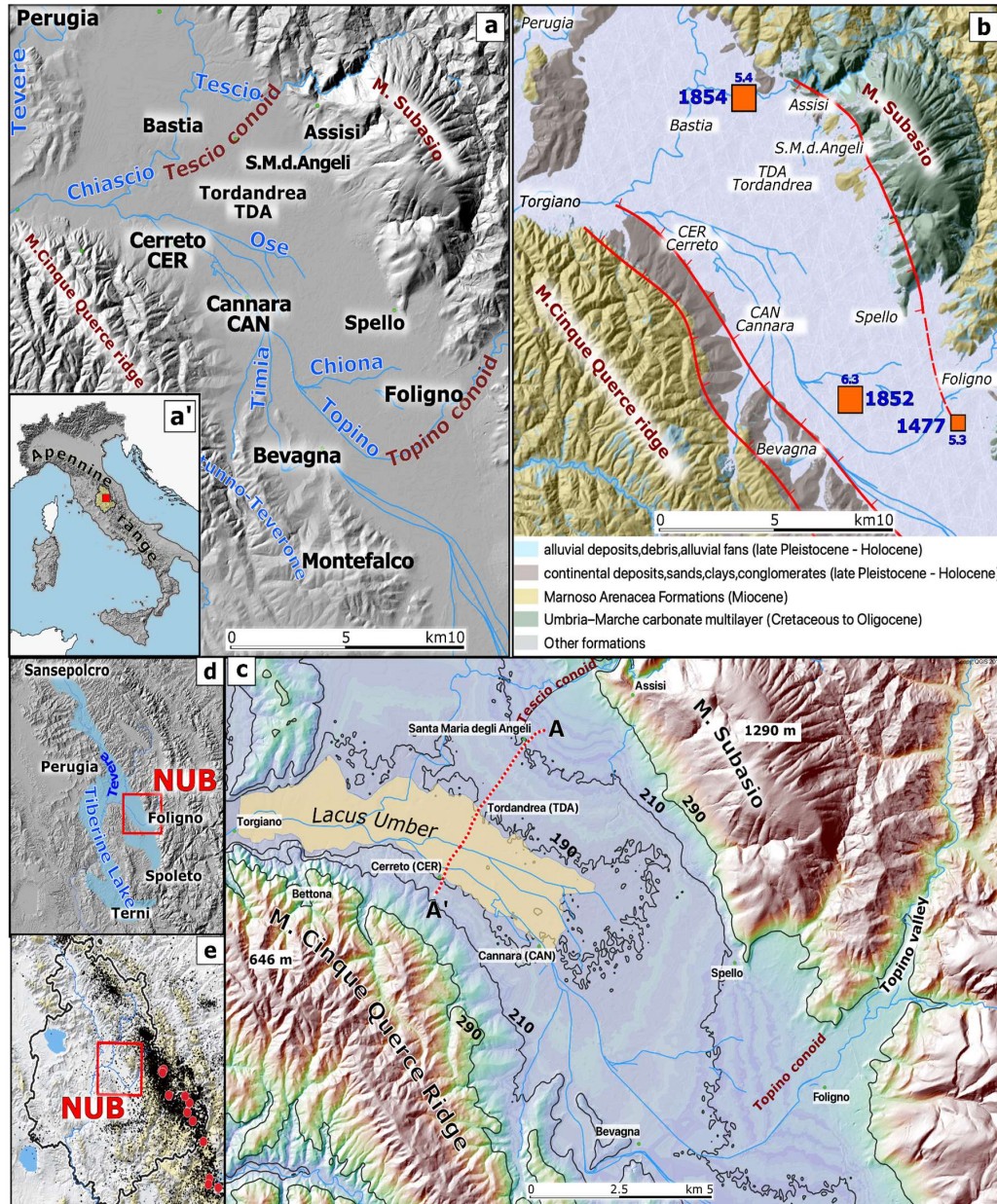

**Figure 2.** (**a**) The principal towns, mountains and rivers of the NUB. (**a′**) The Italian peninsula with the Umbria region (light yellow polygon); the red dot locates the NUB. (**b**) Geology map of the NUB area: red lines are the normal fault traces (the vector layers were provided by Regione Umbria WEBGIS service), and the shaded relief in the background is derived from TINIT DTM [29]; toponyms and river vectors were provided by the Italian ISTAT portal (https://www.istat.it, accessed on 30 March 2023). The orange box locates the epicenters of three main historical earthquakes (Rovida et al., 2021) with the associated magnitude and year of occurrence. (**c**) The NUB orography: the black lines are the isolines showing the valley asymmetry and the shape of the alluvial conoids; the light yellow area locates the sector with the valley's lower elevations, which was occupied by a large lake and marshy area called "Lacus Umber" [30]. The A-A′ red dots locate the topographic and PS velocity profiles of Figure 7. (**d**) The Tevere Valley, along with the northern and southern NUB, formed the Plio-Pleistocene Tiberine Lake (light blue area). (**e**) The Umbria region border and the cloud of small black dots (2.5 < M < 5) and red circles (M ≥ 5.0) are the 1990–2022 seismicity of this sector of the Apennine chain. The coordinate reference system for all the maps in this work is WGS 84/UTM zone 33N.

The NUB is the eastern segment of the Tevere Basin, the former "Tiberine Lake," a Plio-Pleistocene lacustrine basin (Figure 2d) whose sediments that fill the valley were deposited in an environment characterized by braided rivers and shallow lakes [30–34]. The Tevere basin, with an area of about 1800 km$^2$, is the largest of the intermountain basins in the Umbria region (Figure 1). Crossing the entire region from the northern basin of Sansepolcro southward, it divides into two segments near Perugia: the western branch extends southward, in part along the Tevere River, while the eastern one is the NUB [35]. The NUB (Figure 1) is a 180 km$^2$-wide intermontane depression that formed in response to extensional tectonics dissecting the pre-existing fold-and-thrust belt system during the Quaternary period. The Plio-Pleistocene continental sequence infilling the NUB consists of a complex succession of laterally discontinuous deposits, mainly composed of lignitiferous clays, sands, sandy clays and conglomerates. The alluvial sequence is more than 150–200 m thick [36], with lateral variations caused by the presence of fluvial conoids (mainly the Topino and Chiascio alluvial fans, Figure 1) and the structural organization of the underlying bedrock. The Jurassic to Eocene bedrock (overlayed by the Quaternary) to recent formations sequence outcropping in the area, from the youngest to the oldest, is composed of: the recent and older fluvial and alluvial deposits with fine sand and silt levels, lenses of coarse gravel, and sand locally eroded by fluvial terraces [37]. Next is the continental sequence of depositional environments characterized by braided rivers and shallow lakes (Bevagna Unit in Figure 2b) [30–38] outcropping extensively in the south-eastern limits of the NUB [36] (https://dati.regione.umbria.it/dataset/carta-geologica-dell-umbria, accessed on 30 March 2023) [38], followed by the Marnoso Arenacea Formation (early to middle Miocene) [39] and the Cretaceous to Oligocene Umbria–Marche carbonate multilayer of pelagic origin with, at the bottom, the limestones formed on shallow-water platforms in the lower Liassic (Calcare Massiccio Formation) [4,40]. The Upper Miocene Marnoso Arenacea formation (Figure 2b) outcrops in the M. Cinque Querce Ridge (SW of NUB), at the confluence of the Chiascio and Tevere, and on the northern side of the study area near the Torgiano village. Moreover, the Marnoso-Arenacea formation is present along Mt. Subasio's slopes and constitutes the bed of the valley's alluvial and lacustrine deposits.

The Mt. Subasio ridge is dominated by the Umbria-Marche carbonate multilayer (Northern Appenninic Meso-Cenozoic limestone successions). The area has been affected by extensional tectonics since the upper Pliocene—early Pleistocene [41–43], and the 0.5 mm/year of regional uplift competes with the regional subsidence due to the activity of the local normal faults producing subsiding basins incised by modern rivers [44]. The principal NUB watercourses are the Chiascio (length 82 km, 19 m$^3$/s averaged discharge, drainage area 1843 km$^2$), Topino (length 76.6 km, 10.5 m$^3$/s averaged discharge, drainage area 1031 km$^2$) and Clitunno (length 59.3 km, 3.6 m$^3$/s averaged discharge, drainage area 701.4 km$^2$), and the minor ones are the Tescio, Ose, Chiona and Marroggia-Teverone-Timia streams (Figure 2a).

The Topino River flows in the NUB from the Topino Valley (from NE toward SW), then rotates 90° toward NW and, after receiving the waters of the Marroggia-Teverone-Timia and Chiona streams and their tributaries, flows northward along straight and narrow artificial banks (in part parallel to the tributaries), then flows into the Chiascio River near the confluence of the Tevere River. Both the Chiascio and Topino rivers initially flow from north to south and, after crossing the valley, intercept the water of the Clitunno and Marroggia flowing SE before rotating toward NW and flowing to the south-eastern side of the valley. In the NUB, conoids are only present on the western side of the valley because of the greater size of the catchment areas of rivers coming from the Apennines and the presence of a segmented system of both ENE-dipping with prevalent activity and the WSW-dipping active normal faults along the SE and NW sides of the NUB, with SE-NW striking, dissecting the pre-existing (Late Miocene) compressional structures of the Umbria-Marche Apennines (Figure 2c) [45]. This fault system is visible along the Mt. Subasio and Mt. Cinque Querce ridge slopes, forming the NUB graben, and is probably also hidden under the Quaternary sediments filling the NUB [46]. The activity of

these structures is also suggested by the distribution of both instrumental and historical seismicity (Figure 1b–e) [47]. Among the long sequence of earthquakes that struck the valley (CPTI15—https://emidius.mi.ingv.it/CPTI15-DBMI15/, accessed on 30 March 2023), this area experienced two major earthquakes in the 19th century with the epicenters localized here: the 13 January 1832, ME = 6.3, I0 = X, and the 12 February 1854, ME = 5.6, I0 = VII (Figure 2b). In recent times, the earthquake sequences affecting the Apennine sector east of the NUB and the Monte Cinque Querce Ridge (Figure 1e) hit the valley and the urbanized areas, such as on 26 September 1997 (Mw 6.0), often causing considerable damage to civil and industrial structures as well as to technological networks, mainly due to local amplification effects [46].

The NUB is characterized by diffuse and intense agricultural activities, and the fields of wells extracting water from confined aquifers are localized at the confluence area of all the NUB rivers, the principal depocenter of the NUB, the NW flat sector of the valley (Tordandrea-TDA area, Figure 2d). This is a flat area that, up to the 17th century, may have hosted the basin known in the literature as Lacus Persius (Persius Lake) [48] or the more extended Lacus Umber (Umber Lake, Figure 2c) [34]. For the last 30 years, this area has experienced the subsidence phenomena described in the following paragraphs.

## 3. Materials and Methods

By using well-established multi-temporal interferometric techniques, the whole ESA dataset of SAR images covering the Italian territory acquired by ERS, ENV and CSK satellites relative to the 1992–2000, 2002–2010 and 2011–2016 time intervals, respectively, was processed in the framework of the former Italian Ministry of Environment, Land and Sea PST-A project (MATTM, http://www.pcn.minambiente.it/, accessed on 30 March 2023). The PS products of the PST-A project are freely distributed by the Ministry of Environment and Energy Security as shapefiles (points) containing time series data of ground displacement values for both ascending and descending satellite lines of sight (LOS) and deformation rate (Figure 1a). The ERS images (1992–2000) were undertaken by Tele Rilevamento Europa Srl (TRE) using the PSInSAR technique [49], and the ENV data (2002–2010) were processed by eGEOS with the PSPDIFSAR approach [50] The CSK image data (2011–2014) were processed with the PSP-IFSAR analysis algorithm [51–54], but the spatial coverage of the CSK dataset covers only a few areas (Figure 1a). The 2014–2016 SNT PS database covering a wide sector of the central Italy peninsula (Figure 1a) was provided by the EU-funded ESA MEPHYS project and processed by ALTAMIRA-TRE using the SqueeSAR technique [54] and by exploiting the PSI Post-Proc tool, available on the GEP processing services (Geohazards Tep web portal: https://geohazards-tep.eu, accessed on 30 March 2023). The GEP processing services made it possible to calculate the averaged accelerations associated with each PS in the scenes (Tables 1 and 2) and to decompose the LOS deformation into the Up-Down (UD) and East-West (EW) components.

The analysis is focused on the flat sector of the NUB, where the velocity maps show negative ground velocities (i.e., the ground surface moves away from the satellite) during 1992–2016 on the northern side of the valley (Figure 3). Notwithstanding the time gaps between the four satellite acquisitions (Table 1), for the whole observation period, a linear trend of the ground vertical deformation is assumed [5,12]. Furthermore, to compare different satellite measurements, the CSK and SNT LOS time series were re-projected onto the ERS and ENV LOS [10]. Then, each ERS, ENV, CSK and SNT LOS mean velocity map, both on ascending and descending tracks, was post-processed and compared to quantitatively assess the reliability of the datasets. The comparison between the ascending and descending tracks within the region of interest shows good correlations for the ERS, ENV and CSK ($R^2$ = 0.95, 0.93 and 0.84, respectively, Figure 4). These values indicate that the subsidence patterns obtained from the three datasets are dominated by vertical motion. On the other hand, the SNT datasets show a complex distribution of values, probably due to the presence of the East-West slow component of the ground movement. However, for

values characterized by a velocity greater than −10 mm/yr, a continuity in the vertical displacement in both the ascending and descending datasets is visible.

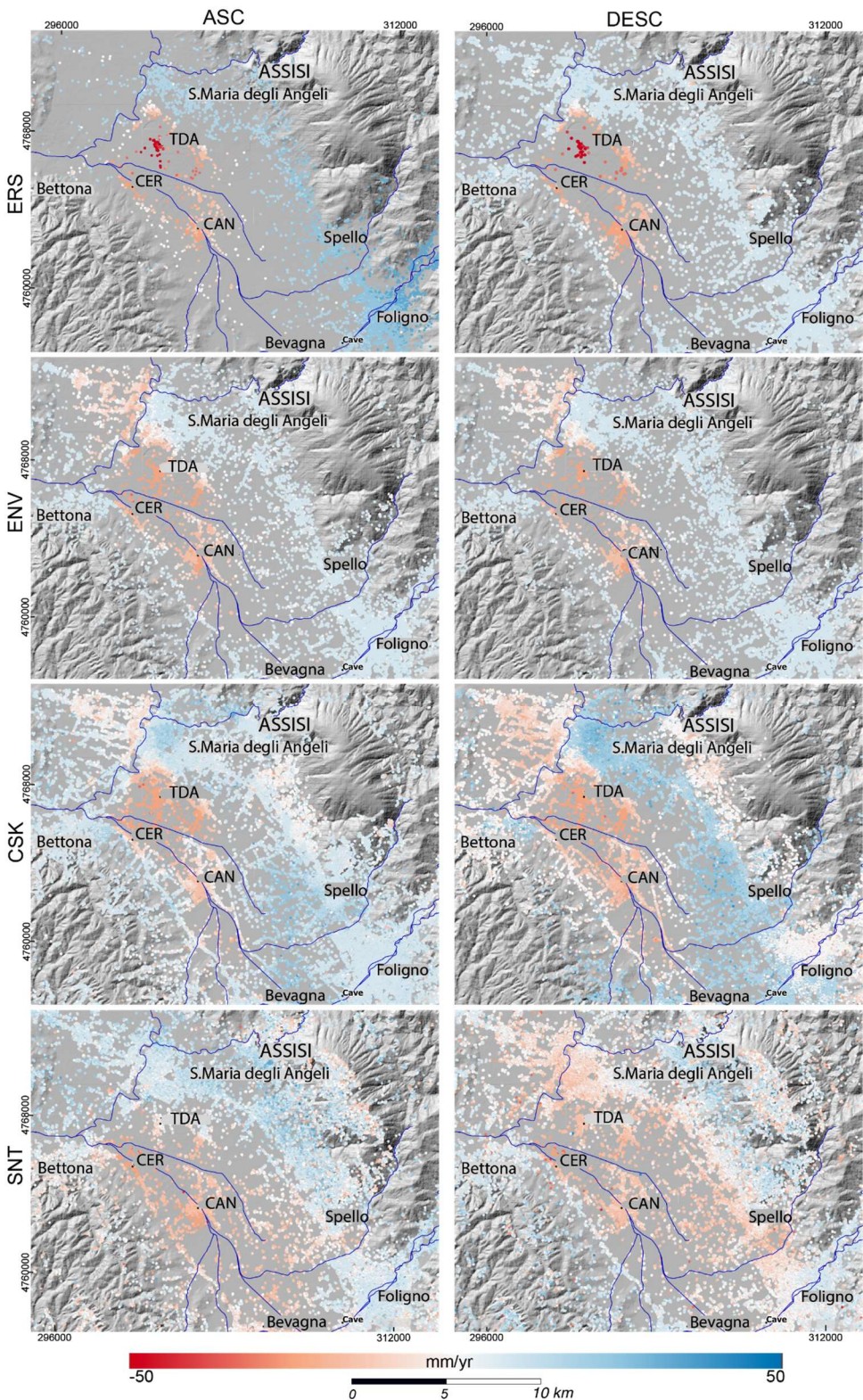

**Figure 3.** Velocity maps in the LOS direction for each of the PS datasets were retrieved from the independent processing of the ascending (**left** panels) and descending (**right** panels) SAR image stacks.

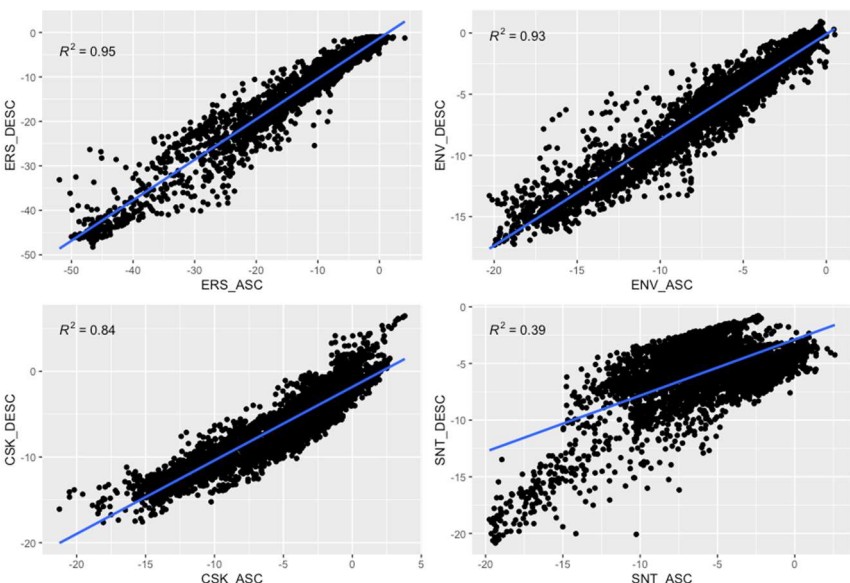

**Figure 4.** For each satellite, the ground velocities in the subsiding area are compared by plotting the common ascending and descending pixel values in the horizontal and vertical axes, respectively. The values are expressed in mm/yr. Black points are the velocities of the whole subsiding study area. The blue line is the linear interpolation, and the R2 values are reported. The ground velocities for ERS, ENV and CSK are comparable for both ascending and descending orbits with R2 close to 1, while the SNT velocities show values more dispersed (with a much lower R2) than in the previous periods.

It is worth pointing out that in this work we have focused our attention only on vertical displacements; other (East-West) displacements and minor signals were not taken into account. Three NUB sites were selected to analyze the ground displacement: the Tordandrea (TDA) and Cerreto (CER) areas with the highest deformation rates and Cannara (CAN) with intermediate behavior in terms of ground velocities (Figures 2a and 3). The ERS, ENV and CSK ascending and descending datasets were combined to retrieve the vertical (Up) and East-West (East) velocity fields (Figure 5) [10,52]. By using 100 m square elements to resample to a regular grid, the PS point datasets from both tracks were obtained. The Up and East values were achieved by applying Equation 1 and assuming null or negligible North–South velocity because of the quasi-polar orbit of the satellite and the right lateral observation geometry [53,54].

$$Up = \frac{V_d sin\theta_a - V_a sin\theta_a}{sin(\theta_a + \theta_d)} \quad East = \frac{V_d cos\theta_a - V_a cos\theta_d}{sin(\theta_a + \theta_d)} \tag{1}$$

The observed vertical velocities (Figure 5, Table 4) show that for the ERS dataset (1992–2000), the highest vertical velocity is around 51 mm/yr with a total ground displacement of around −445 mm in the TDA area. For the ENV data (2002–2010), the maximum velocity value is moving SW toward the CER area, decreasing to −20.1 mm/yr with a displacement of −152 mm. In the same area in the 2011–2014 (CSK) and 2014–2016 (SNT) periods, the ground velocities are, respectively, −15 mm/year (with a total displacement of around −116 mm) and −16 mm/year (with a total displacement of around −47 mm). In the CAN area, the vertical velocity decreased between 1992 and 2010 (ERS, ENV and CSK acquisitions), stabilizing during the period of the SNT acquisition (Table 4). The cumulated displacement, i.e., in the TDA area, is 6203 mm, but the value does not take into account the satellite acquisition gaps.

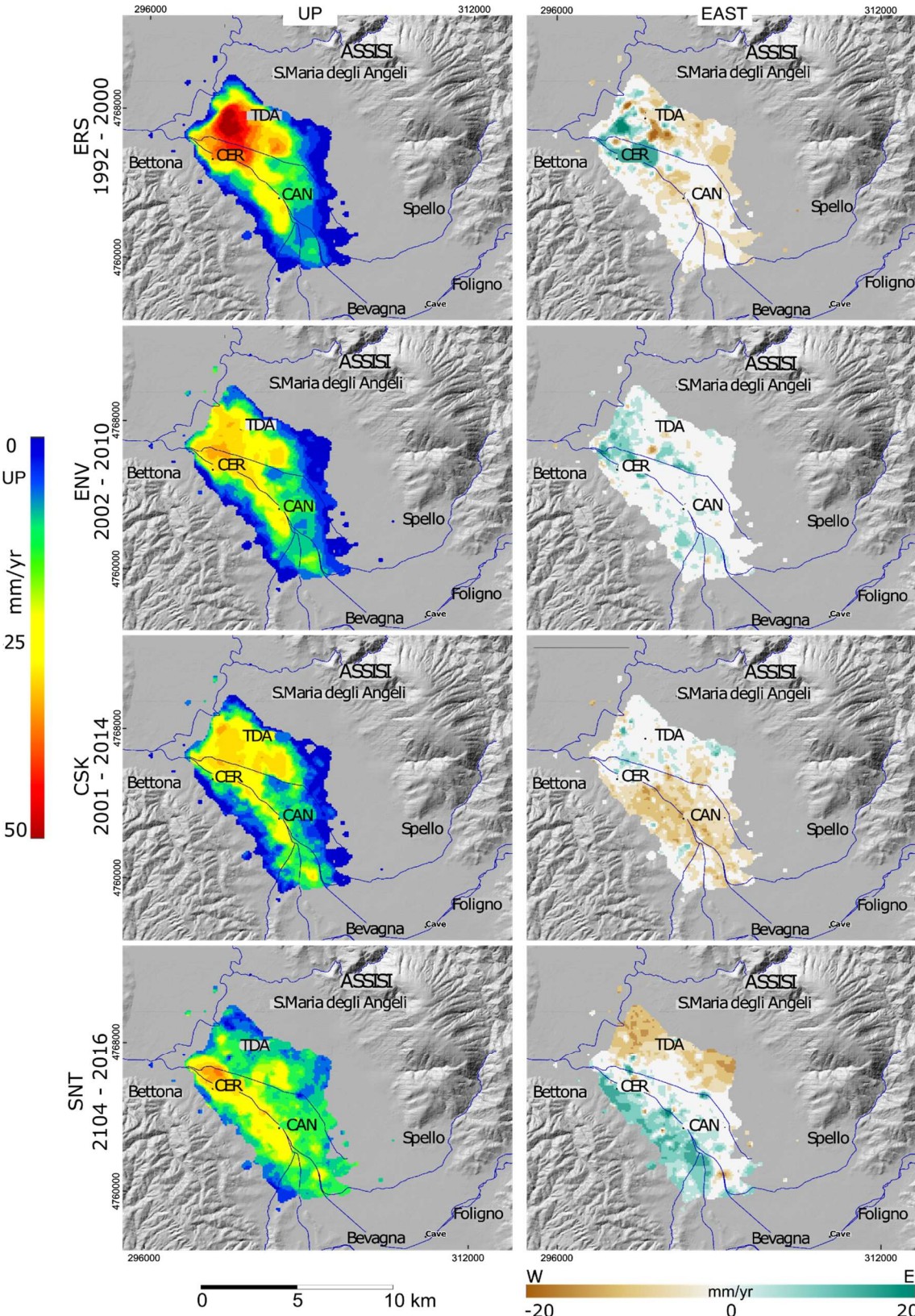

**Figure 5.** Velocity maps for Up (vertical displacements in the left column) and East-West (horizontal displacements in the right column) velocity components (mm/yr) in the NUB area during the four time intervals.

**Table 4.** The ground velocities and vertical ground displacements in three selected areas (CAN, CER and TDA), measured by four satellites (ERS, ENV, CSK and SNT).

| | ERS 1992–2000 | | ENV 2002–2010 | | CSK 2011–2014 | | SNT 2014–2016 | | Total Displacement |
|---|---|---|---|---|---|---|---|---|---|
| Years | 8.74 | | 7.56 | | 2.94 | | 1.86 | | |
| | mm yr | mm | mm yr | mm | mm yr | mm | mm yr | mm | mm |
| TDA | −50.9 | −444.9 | −15.3 | −115.7 | −16.2 | −47.6 | −6.5 | −12.1 | −620.3 |
| CER | −33.2 | −290.2 | −20.1 | −152.0 | −19.7 | −57.9 | −23.0 | −42.8 | −542.8 |
| CAN | −15.9 | −139.0 | −13.4 | −101.3 | −10.7 | −31.5 | −11.3 | −21.0 | −292.7 |

## 4. Discussion

The subsidence observed in the NUB is analyzed by exploiting available datasets from MInSAR products. Moreover, the observed deformation trend during the 1996–2002 time interval is compared with the piezometric level time series measured in some of the wells of the same area (in particular the P9, P16 and P26 wells, Figure 6d). Although the SAR acquisitions of the four exploited satellites (ERS, ENV, CSK and SNT, Table 2) do not have time cover continuity to directly extrapolate measures of ground displacement, the time gaps were filled by applying a linear fit to the different displacement intervals to obtain a continuous measure of the deformation (Figure 6a–c). The subsidence calculated using the available PS values in the TDA area is up to −70 cm (Figure 6d). After the integration of the four datasets, the deformation series were compared with local groundwater level time series (Figure 6a–c; the well's water level measurements are provided by Arpa Umbria, https://apps.arpa.umbria.it/acqua/contenuto/Livelli-Di-Falda, accessed on 30 March 2023). Since 1990, the artesian aquifer localized in the northern NUB, known in the bibliography as the "Cannara aquifer" (ARPA 2007—https://www.arpa.umbria.it/resources/docs/ACQUIFERI_valle_umbra.pdf. In Italian. Accessed on 30 March 2023) [55], has undergone intense pumping for drinking water uses. Historical data show that at the beginning of the 1970s, the piezometric level in this area was about 10 m above ground level (around 198 m.a.s.l.). Later on, at the end of the 1980s, the groundwater level was around 188–190 m a.s.l.

The start of operations that used water wells, with increasing withdrawals up to 250 l/s, led to a significant depression of the aquifer. In the period 1990–2002, there was a decrease in the height of water in the wells of the observed subsiding TDA area, while the data available for the following period (2005–2020) show a positive trend in the level of water, in particular in the well P26 (Figure 6a). In the CAN area (Figure 6c), as in the TDA area, the observed 1992–2016 subsidence is accompanied by a positive trend in the P16 well water levels. On the contrary, in the subsiding CER area, the negative trend of the 2004–2019 water levels in the P9 well and the continuous lowering of the ground surface are well correlated (Figure 6b). In summary, the piezometer of the P26 shows that the water level has increased by 7 m on average during 2006–2016. The well's water fluctuations are also visible due to seasonal intermittent water withdrawals for agricultural purposes in combination with the water inputs due to rainfalls; these oscillations are also visible in the deformation time series with reduced dynamics. The deceleration of the ground velocity in the P26 surroundings can be interpreted as a consequence of the deep aquifer compaction [25,55] due to excessive water withdrawal. In the P16, the average water level shows a positive trend between 2006 and 2016, and the ground subsidence trend is constant. A different pattern of behavior can be observed in P9, where both the water level and the soil's lowering trend have a negative sign.

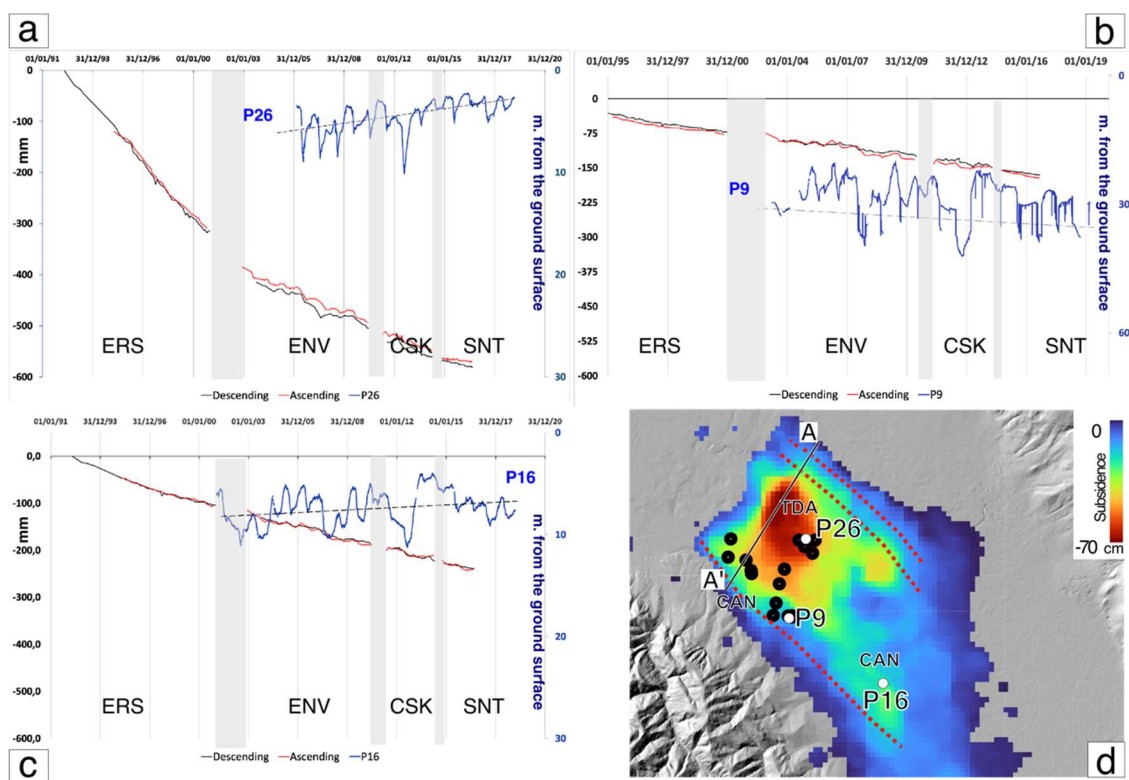

**Figure 6.** (**a–c**) Ground displacements measured by satellites in the descending (black lines) and ascending (red lines) LOS expressed in mm in three selected zones in correspondence with P9, P16 and P26 wells. The PS time series were compared to the water levels (the blue lines are the water level in the wells from the surface expressed in meters): (**a**) Tordandrea (TDA) and P26 water levels; (**b**) Cerreto (CER) and P9 water levels; (**c**) Cannara (CAN) and P16 water levels. (**d**) Total subsidence map measured by ERS, ENV, CSK and SNT satellites between 1992 and 2016, where black circles between TDA and CER are wells from the Cannara well fields and the white circles are the P9, P16 and P26 wells. The A-A′ line is the profile of Figure 7. The inferred fault system is traced with red dashed lines (see Figures 7 and 8).

In February 2019, using the Differential Global Positioning System (DGPS) technique, a topographic campaign produced a detailed elevation profile crossing the TDA area (Figure 7) from the stable area on the eastern side of the valley (S. Maria Degli Angeli village) through TDA and toward the CER area. Along the profile, the maximum elevation difference is −33 m between stable areas and the subsiding zone (TDA), with two steps correlated to the changes of ERS, ENV, CSK and SNT ground velocities measured along the same elevation profile trace (Figure 7). The subsidence velocity changes that were measured along the profile in Figure 7 are correlated to the plano-altimetric variations along the topographic profile.

This suggests that the deposits react to the depth reduction in deep aquifer volumes in proportion to their thickness and, therefore, to changes in bedrock depth (Figure 8). This would confirm the already hypothesized presence of buried fault scarps parallel to what was observed on the surface along the reliefs that border the valley to the NE and SW (Figure 2a). The valley bedrock organization according to the graben style is described in [46]. What is tricky to identify through this study is the activity of the fault system due to the presence of overlapping movement signals recorded by the satellite time series, making it difficult to recognize the tectonic signature from the surface displacements due to the water variations in the deep aquifers. Anyway, the presence of historical earthquakes and hypocenters located in this sector of the Apennine chain (Figure 2b,e) indicates the presence of tectonic activity with displacements along this fault system.

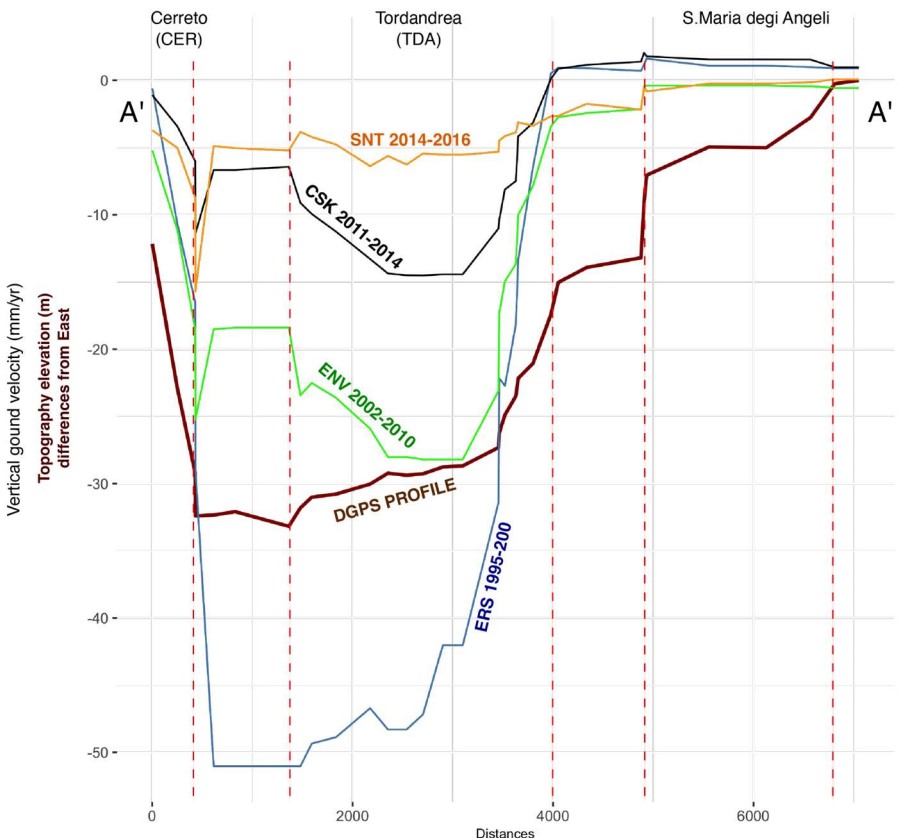

**Figure 7.** (Upper panel) Topographic and velocity profiles: the brown line is the elevation profile of the valley along A-A' (Figures 3c and 6d), and the blue, green, black and orange lines are the ERS, ENV, CSK and STN satellites' ground vertical velocities profiles, respectively, along the same profile. In the Y-axis, the values express both the mm/year of vertical velocities of ground displacement and the height differences, in meters, measured along the profile A-A' with respect to the initial point A. The red dashed lines (see Figure 6d) identify the orographic steps along the profile that are correlated to the changes in ERS, ENV, CSK and SNT ground velocities measured along the same elevation profile trace (horizontal vs. vertical scale is 1:100).

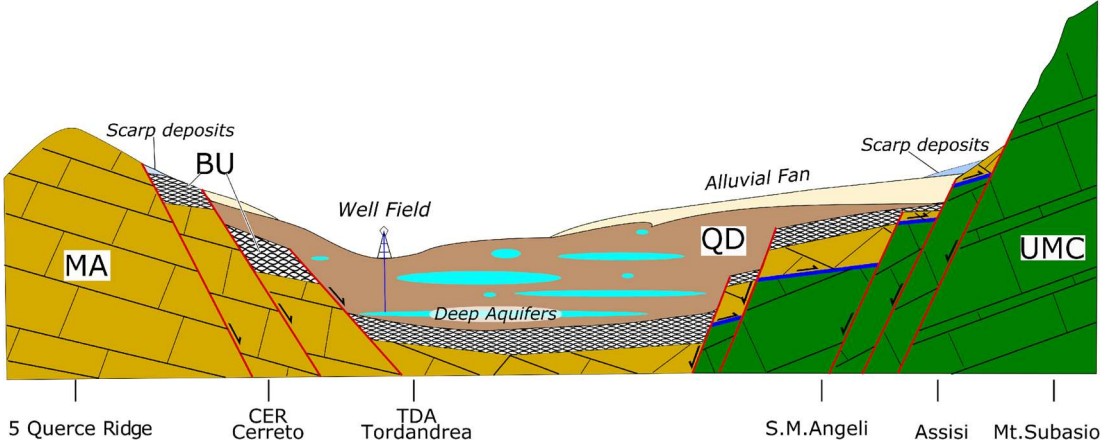

**Figure 8.** Schematic and out-of-scale geological profile along the A-A' trace (mod. [46]): (MA) Marnoso Arenacea Formation; (UMC) North Appeninic carbonate multilayer; (BU) Bevagna unit; (QD) Quaternary deposits.

### 5. Conclusions

During the last few decades, the availability of very large datasets of SAR satellite acquisitions and new technologies based on advanced cloud computing, implementing the multi-temporal differential interferometry (MDInSAR) technique, have allowed the observation of the earth deformation phenomena with unprecedented time and spatial extensions in the framework of the Earth Observation Data.

To take advantage of the opportunities offered by the availability of long-time series of satellite data and online advanced data processing technologies, here we used the SAR images acquired by SAT time-series elaborated within the ESA-MENPHYS (Sentinel 1) and the Italian projects MATTM-PST-A to produce time-series of ground displacement PS datasets. The ground subsidence that occurred in Central Italy is analyzed in temporal and geometric terms by exploiting the 1992–2016 datasets of the ESA-MENPHYS and MATTM-PST-A MInSAR PS products. The ground subsidence was correlated with the intense artificial water withdrawal from the deep local aquifer, and the shape of the deformation gave information about the shapes of bedrock geometries hidden under the valley deposits. Ref. [46] confirmed that under the valley sediments is present the NW-SE trending fault system with SE dipping faults dislocating the northern-eastern lithologies of the Mt. Subasio and in the NW dipping of the Cinque Querce ridge (Figure 1). The over-pumping from the confined groundwater of the Cannara aquifer causes the irreversible collapse of the aquifer, highlighted by the ground surface subsidence that is up to 70 cm in the TDA area (Table 4), with a consequent decrease in the availability of water for drinking, agricultural and industrial purposes [55]. Indeed, the Cannara well field is characterized by a reservoir with a withdrawable flow rate greater than 30 L/s that is not currently substitutable with other supply sources, also considering the impact of climate change on the groundwater resources. In addition, significant historical and instrumental seismicity suggests that these faults are still active and responsible for the area's seismic activity.

The main results of this work are: (1) the fruitful use of the already elaborated and available PS time series provided by Italian and European-funded projects that widely circulate within the scientific community and which can be exploited for various purposes and for the study of numerous natural and anthropogenic phenomena, as in the case described in this work; and (2) the correlation between the spatial shape and evolution of the valley deposits subsidence with the presence of hidden faults according to the extensional graben style. The analysis of more recent data, such as the SNT time series, will confirm the decrease in the subsidence phenomenon and, when added to the data presented here, could help with the description of the tectonic activity present in the Northern Umbria Valley.

**Author Contributions:** C.A.B. and F.M. have equally contributed to the paper. All authors have read and agreed to the published version of the manuscript.

**Funding:** This research received no external funding.

**Data Availability Statement:** The ERS, ENVISAT and COSMO Sky-Med PS time series are produced and can be required within PST-A "Extraordinary Plan for Environmental Remote Sensing"), funded by the former Italian Ministry of Environment, Land and Sea (MATTM)—http://www.pcn.minambiente.it/ (accessed on 30 March 2023).

**Acknowledgments:** We are grateful to the reviewers whose comments and suggestions have helped improve and clarify this manuscript. Thanks to Luca Pizzimenti for the help and availability in the DGPS measures campaign. This work enjoyed the continued support of Maria Teresa, Caterina and Gabriele.

**Conflicts of Interest:** The authors declare no conflict of interest.

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
