# Peer review of "Spatiotemporal Evolution of Ground Subsidence and Extensional Basin Bedrock Organization: An Application of Multitemporal Multi-Satellite SAR Interferometry"

_geosciences, doi:10.3390/geosciences13040105_

Round 1
Reviewer 1 Report
The land deformation is very important topic in all over the world. It is a challenge to observe it based on InSAR for the long time period.
There are also many institution and research group, which they focus on this topic and could be interested in this study.
However, the major issue of the manuscript, as I see it, is that the authors did not present clearly, what is new in this research? What is the aim of the study?
The deformation pattern due to dewatering of aquifer is nothing new.
In addition, I have a few specific comments:
1. In tables 2 informations for SNT are wrong.
2. Table 3 is not necessary.
3. Why only in SNT result was visible horizontal component of deformation? Not for the other sensors. This fact should vaideted in some way.
4. Why 'Conclusion section' has number 2.
5. The figures are illegible.
6. Tables are graphis and they are illegible. It should not be.
7. For what is calculated vertical displacement is later it not use in reserch?
I fully recommend a deep revision of the manuscript and defining the research goal. In order to improve you can consider:
1. Expand analysis for SNT, for which acquisition time after 2016 was reduce to 6 days.
2. Decompose the time series into trend and seasonality. This may allow the correlation of the height of the water table with displacements values from InSAR. This issue is the most interesting part of the manuscript and should definitely be highlighted more.
Author Response
Thank You for your comments and suggestions. We report below the responses
----
The deformation pattern due to dewatering of aquifer is nothing new.
Response: We add and rewrite few lines in the introduction and in the conclusions to highlight more clearly the major issue and the result of the study. The paper focuses on the correlation between the spatiotemporal evolution of the subsidence with the geometries (shapes and extension) of the hidden structures buried under the valley's recent deposits.
In addition, I have a few specific comments:
- In tables 2 informations for SNT are wrong.
Response 1: Done. The compile error has been fixed.
- Table 3 is not necessary.
Response 2: Since we use the freely available PS database provided by the Italian Ministry, we indicate in Tab 3 and in Figure 1 the name of the shape file analysed for this work in order to facilitate the reproducibility of the measurements. We suppose this is useful for the readers.
- Why only in SNT result was visible horizontal component of deformation? Not for the other sensors. This fact should vaideted in some way.
Response 3: Since we detected the presence of horizontal components in all the time series, in the fourth paragraph of the Materials and Methods chapter, we declare that in this work we decided to focus our attention only on the vertical displacements useful for the identification of the relative thicknesses of the deposits and, therefore, of the presence of bruised lateral variations of the bedrock shapes. For this reason, the horizontal displacements (East-West) and the minor signals are not taken into consideration in this work, however they are presented for completeness.
- Why 'Conclusion section' has number 2.
Response 4: Done.
- The figures are illegible.
Response 5: We correct the text's dimension and make other enhancements to improve the quality of the figures.
- Tables are graphis and they are illegible. It should not be.
Response 6: We insert the Table format instead of the graphic format
- For what is calculated vertical displacement is later it not use in reserch?
Response 7: In Figure 7 the lateral variations of vertical displacements for each of the satellite measurements (and time intervals)were correlated to the lateral variations of the recent deposits and, as a consequence, the presence of vertical discontinuity on the buried bedrock is referable to probable fault scarps. In this context, the analysis of the data derived from the extraction of the vertical components from the PS time series is purely qualitative.
I fully recommend a deep revision of the manuscript and defining the research goal. In order to improve you can consider:
- Expand analysis for SNT, for which acquisition time after 2016 was reduced to 6 days.
- Decompose the time series into trend and seasonality. This may allow the correlation of the height of the water table with displacements values from InSAR. This issue is the most interesting part of the manuscript and should definitely be highlighted more.
Response: The latter points are a tip to a very interesting analysis of the data and natural phenomena induced by human activity for another more complete and detailed work. However, the objective of this work is the analysis of the already elaborated time series available of PS provided by two projects (PST Italian Ministry and MEMPHIS EU Project). This kind of data set widely circulates within the scientific community and we believe it is useful to understand their full potential by using them for various purposes. For this reason, we do not process "new" SNT acquisitions. Moreover, we add the underlined line in the final part of the Introduction chapter to improve the comprehension of one of the scopes of the article.
Reviewer 2 Report
This paper uses InSAR from multiple satellite missions to pool together a deformation time series spanning 24 years over the Northern Valle Umbra Basin in Italy. The authors show the basin to subsiding over this time period and loosely attribute the cause to buried fault activity in the basin.
My main comments are:
1. The paper spends a lot of space at the beginning talking about big data. But from what I read the actual work does not involve any innovative big data analysis. Most of the analysis in the paper is now routine for InSAR data. I think the focus instead should probably remain on the issues regarding subsidence.
2. There is no mention of the spatial reference for each image. This needs to be clearly stated before comparison between different satellite sensors can be made. But I am happy to see the conversion to the Envisat line of sight before comparison. This is great. But please briefly explain how you did this.
3. The location of table 1 confuses the narrative, specifically the cluster IDs. Consider removing it or incorporating the information into a later table
4. I am not convinced by your deconvolved vertical InSAR velocities in Figure 5. The ERS, ENVISAT AND CSK show up to 50mm/yr uplift at the periphery of the subsidence bulge. Surely this is incorrect. I suspect this is an issue related to the spatial reference. Given this, I don’t know how you don’t have any uplift in Figure 6d. Can you explain?
5. I am also not convinced about the main argument in your paper where the subsidence patterns are controlled by fault activity. You only really discuss this in one very brief paragraph in the discussion. I would like to see much more detail on this with more convincing data and analysis. Could you plot the faults on your maps for example and show the spatial relationship between subsidence and fault locations? This is potentially very interesting if true.
Author Response
Thank You for your comments and suggestions. We report below the responses to the comments
------
My main comments are:
- The paper spends a lot of space at the beginning talking about big data. But from what I read the actual work does not involve any innovative big data analysis. Most of the analysis in the paper is now routine for InSAR data. I think the focus instead should probably remain on the issues regarding subsidence.
Response 1: Actually, the reference to Big Data at the beginning of the article can cause a misunderstanding in the aims of the work itself. Following this suggestion, we eliminate the lines that contain the concept of Big Data
- There is no mention of the spatial reference for each image. This needs to be clearly stated before comparison between different satellite sensors can be made. But I am happy to see the conversion to the Envisat line of sight before comparison. This is great. But please briefly explain how you did this.
Response 2: - As far as spatial references are concerned, they will be inserted for each image. The CSK and SNT PS displacements were projected onto the ERS and ENV LOS under the assumption that the deformation is mainly vertical, neglecting the east-west and the north-south components of the deformation, hence using the same approach described in Murgia et al. 2019 https://doi.org/10.3390/rs11192246 (paragraph “4.2. Deformation Time-Series Analysis”)
- The location of table 1 confuses the narrative, specifically the cluster IDs. Consider removing it or incorporating the information into a later table.
Response 3: We agree, and Tab 1 has been removed and the caption has been rephrased.
- I am not convinced by your deconvolved vertical InSAR velocities in Figure 5. The ERS, ENVISAT AND CSK show up to 50mm/yr uplift at the periphery of the subsidence bulge. Surely this is incorrect. I suspect this is an issue related to the spatial reference. Given this, I don’t know how you don’t have any uplift in Figure 6d. Can you explain?
Response 4: Actually it is likely that the uplift at the periphery of the subsidence is related to the spatial reference, however the type of legend used saturates the values close to 0. But the purpose of the paper is qualitative analysis rather than quantitative and the uplift areas are not considered because they are useless for identifying variations in the thickness of the sediments. The velocity changes value increases laterally with steps in correspondence with lateral topographic variations (Fig 7) and with discontinuities of the bedrock identified by Famiani et al. (2020) and therefore with probably buried fault scarps
- I am also not convinced about the main argument in your paper where the subsidence patterns are controlled by fault activity. You only really discuss this in one very brief paragraph in the discussion. I would like to see much more detail on this with more convincing data and analysis. Could you plot the faults on your maps for example and show the spatial relationship between subsidence and fault locations? This is potentially very interesting if true.
Response 5: We add more considerations in the discussion paragraph and we trace t the projection of the hidden faults in the map of Figure 6.
Reviewer 3 Report
The article is good and in the trend of the investigation of deformation of the surface and subsidence. I think it can be publish after some redaction on comments below:
1. Please, separate out the used data and method from Introduction, doing it shorter and clear; the discussed data could be added to special section 3.
2. Take attention to “a)” in signature to the Table 1. If the “a)” is referred to figure 1 write it clear, please.
3. Please, move signature of profile A’-A in Figure 2c to “c)”.
4. On Figure 2 “b” and “c” could be combined. In the whole figure 2 could be re-organized.
5. Too long signature to Table 2. Please move part in the manuscript body or in the footnote.
6. Figure 5. Indicate years inside figures and remove it from signature. There are not “a, b, c, and d” in the Figure 5.
7. Conclusion should be re-organized, clear and short. You don’t need to do extended abstract of your paper or discussion. You need designating the main result points of your research.
Author Response
- Please, separate out the used data and method from Introduction, doing it shorter and clear; the discussed data could be added to special section 3.
Response 1: The description of the data in the Introduction chapter has been removed because redundant with the description present in the Material and Methods chapter.
- Take attention to “a)” in signature to the Table 1. If the “a)” is referred to figure 1 write it clear, please.
Response 2: The table 1 has been removed and the caption has been rephrased.
- Please, move signature of profile A’-A in Figure 2c to “c)”.
Response 3: Hoping we interpreted the comment correctly, trace A-A' of figure 2c is in the caption after -> "c)"
- On Figure 2 “b” and “c” could be combined. In the whole figure 2 could be re-organized.
Response 4: Actually, in the first version of the figure, 2b, 2c and 2e were merged, but the result was, in our opinion, confusing, so we decided to separate the subjects of geology and paleoseismology, now in 2b, from topography and paleogeography ( "Lacus Umber") in 2c, and from the epicenters of the last 30 years (now in 2e). Given the appearance and the previous organization, we would prefer to leave the figure of the general framework organized like this because we think that the proposed themes are more decipherable.
- Too long signature to Table 2. Please move part in the manuscript body or in the footnote.
Response 5: We agree. The information is already present in the text and therefore redundant. Removed from the caption.
- Figure 5. Indicate years inside figures and remove it from signature. There are not “a, b, c, and d” in the Figure 5.
Response 6: The years are now indicated inside the figure and the caption has been rephrased.
- Conclusion should be re-organized, clear and short. You don’t need to do extended abstract of your paper or discussion. You need designating the main result points of your research.
Response 7: The conclusions have been rephrased according to the suggestions.
Round 2
Reviewer 2 Report
The authors responded positively to my second comment requesting that you show or explain where the spatial references are for each data stack. But I can’t see this change mentioned anywhere in the text or shown in an of the Figures.
Figure 6d. Thank you for showing the fault. But why are there only 3 faults when there are 8 mapped in Figure 7?
Figure 7: I don’t understand how you can show the elevation profile in metres on the same scale as the ground velocity in mm/yr. Please correct this or explain. Please also state the distance units on the x axis. Please make sure the x axis of both the geological and the topography/insar panels are aligned correctly.
Figure 7 caption: You mention that profile A-A’ is shown in Figure 3c. I can’t see this in Figure 3 anywhere. Could you show this profile on Figure 6d instead please.
Author Response
The authors responded positively to my second comment requesting that you show or explain where the spatial references are for each data stack. But I can’t see this change mentioned anywhere in the text or shown in an of the Figures.
Response 1: Yes, It's true, the coordinate grid is already present in the map, but now we also add the line - The coordinate reference system for all the maps in this work is: WGS 84 / UTM zone 33N - in the caption of Figure 2 and as a reference for all figures
Figure 6d. Thank you for showing the fault. But why are there only 3 faults when there are 8 mapped in Figure 7?
Response 2: Here we trace only the position of the hidden fault system inferred from the surface ground deformation shape. The caption of figure 6, in the last part, is rephrased as: “The inferred fault system is traced with red dashed lines (see fig 7 and 8)”
Figure 7: I don’t understand how you can show the elevation profile in metres on the same scale as the ground velocity in mm/yr. Please correct this or explain. Please also state the distance units on the x axis. Please make sure the x axis of both the geological and the topography/insar panels are aligned correctly.
Response 2: In the “upper panel” caption we modify the lines describing the values of the Y-axis in: "In the Y-axis the values express both the mm/year of vertical velocities of ground displacement and the height differences, in meters, measured along the profile A-A' with respect to the initial point A" .
Since in the upper panel the quantities for vertical and horizontal axes are expressed, while in the lower the relationship between the two dimensions is relative, therefore, to avoid confusion, the upper Panel becomes Figure 7 and the lower one Figure 8 and are conveniently moved in the text.
Figure 7 caption: You mention that profile A-A’ is shown in Figure 3c. I can’t see this in Figure 3 anywhere. Could you show this profile on Figure 6d instead please.
Response 3: La figura 6d è stata corretta aggiungendo la traccia del profilo A-A'